# Subcritical Water Extraction as an Effective Technique for the Isolation of Phenolic Compounds of *Achillea* Species

Katarina Radovanović [1], Neda Gavarić [1], Jaroslava Švarc-Gajić [2], Tanja Brezo-Borjan [2], Bojan Zlatković [3], Biljana Lončar [2] and Milica Aćimović [4,*]

1 Department of Pharmacy, Faculty of Medicine, University of Novi Sad, Hajduk Veljkova 3, 21000 Novi Sad, Serbia
2 Faculty of Technology Novi Sad, University of Novi Sad, Bulevar Cara Lazara 1, 21000 Novi Sad, Serbia
3 Department of Biology and Ecology, Faculty of Sciences and Mathematics, University of Niš, Višegradska 33, 18000 Niš, Serbia
4 Institute of Field and Vegetable Crops Novi Sad, National Institute of the Republic of Serbia, Maksima Gorkog 30, 21000 Novi Sad, Serbia
* Correspondence: acimovicbabicmilica@gmail.com

**Abstract:** The genus *Achillea* has significant medical potential due to the presence of highly bioactive compounds in its chemical composition. To take advantage of plants' biomedical potential, it is of great importance to use a proper extraction process. This study aimed to determine and compare the preliminary chemical composition of five different *Achillea* species extracted with two conventional (infusion and maceration) and two non-conventional (ultrasound-assisted extraction (UAE) and Subcritical water extraction (SWE)) techniques. The extracts were prepared using the previously described procedures for infusion, maceration, UAE and SWE extraction. For all extracts, the extraction yield (dry extract (DE)) was determined. The analyzed extracts were preliminarily chemically characterized spectrophotometrically in terms of total phenolic content (TPC) and total flavonoid content (TFC). The obtained results showed that non-conventional techniques delivered higher values of TPC and TFC than conventional. There is a statistically significant increase in DE and TPC content when applying SWE for all observed *Achillea* species. The highest DE value, $48.80 \pm 1.76\%$, was observed for *A. asplenifolia*. The highest TPC values were observed after applying SWE: $93.63 \pm 1.01$ mg GAE/g DE for *A. millefolium*, and $90.12 \pm 0.87$ mg GAE/g DE for *A. crithmifolia*. The results for TFC revealed a statistically significant difference in values, with *A. nobilis* subsp. *nelreichii* as the sample with the highest content of TFC ($11.11 \pm 0.22$ mg QE/g DE) when using UAE. Consequently, it could be concluded that SWE is a superior non-conventional extraction technique, and *A. nobilis* subsp. *nerleichii* presents as the most promising plant.

**Keywords:** yarrow; green extraction techniques; phenolic compounds; principal component analysis

## 1. Introduction

One of the most important genera in the Asteraceae family is the genus *Achillea,* which contains around 130 species spread all over the world, with Serbia being one of its main habitats in Europe. The genus *Achillea* has huge ethnopharmacological value because it includes numerous medicinally important plants. The phytochemistry of this genus is composed of the dominant groups of secondary metabolites, these being monoterpenes, sesquiterpenes, sesquiterpene lactones and proazuelens, as well as flavonoids, these being phenolic glycosides, lignans and tannins. Research revealed that many of these compounds are highly bioactive and that they contribute to the significant medical potential of *Achillea* species [1,2]. The most studied species is *A. millefolium*, the official representative of the genus *Achillea*. Other species of this genus have been poorly studied and literature data is limited, but in folk medicine they are considered to have the same effects as *A. millefolium*.

Due to their wide representation in the flora in Serbia, they are traditionally used as a local substitute for *Achillea millefolium*.

In order to fully benefit from the biomedical potential of a plant, it is of great importance to use a proper extraction process. Final results and outcomes of any study on plants is highly conditioned by the selected extraction method, as the first step of such studies. Extraction techniques are roughly divided into two groups: conventional and non-conventional. Conventional methods are based on extracting power of different solvents with the possibility of convective mass transfer through mixing. The most common classical techniques are Soxhlet extraction and maceration [3]. Extraction efficiency is highly influenced by the solvent, applied technique and the sample matrix. Besides solvent selectivity towards target chemical class and interfering compounds present in the sample matrix, environmental consideration, safety and financial feasibility must be taken into account in the selection of a specific extraction approach for bioactive compound extraction. Because of these demands and limitations, alternative extraction techniques are increasingly being used [3,4].

Non-conventional extraction techniques are promising approaches that overcome the shortcomings of conventional techniques. Some of the most promising techniques are ultrasound-assisted extraction (UAE), enzyme-assisted extraction, microwave-assisted extraction, pulsed-electric-field-assisted extraction, sub- and super-critical fluid extraction and pressurized-liquid extraction. Some of these techniques are considered 'green techniques' since they are in accordance with the standards set by the Environmental Protection Agency, USA. These processes are considered more environmentally friendly due to decreased use of synthetic and organic chemicals, reduced operational time and better yield and quality of extract [5–7].

Subcritical water extraction (SWE) is one of these green extraction technologies, and is the most promising one. The term subcritical water refers to liquid water at temperature and pressure below its critical point (Tc = 374.15 °C, Pc = 22.1 MPa). The pressure of the subcritical water must be higher than the vapor pressure at a given temperature to keep the water in a liquid state. The physico-chemical properties of subcritical water are determined by the temperature. A substantial advantage of SWE is that water is a safe, nontoxic, easily accessible solvent with tunable selectivity and superior efficiency in respect to conventional extraction techniques [8,9].

Therefore, the aim of this study was to determine and compare the preliminary chemical composition of five different *Achillea* species extracted with two conventional (infusion and maceration) and two non-conventional (UAE and SWE) techniques with water as conventional solvent. The utilization of principal component analysis (PCA) enabled differentiation among the analyzed *Achillea* species samples and a graphical illustration of relations between observed preliminary chemical composition parameters.

## 2. Materials and Methods

### 2.1. Plant Material

The aboveground flowering parts of five *Achillea* species (*Achillea millefolium*, *Achillea clypeolata*, *Achillea asplenifolia*, *Achillea nobilis* subsp. *nelreichii* and *Achillea crithmifolia*) were collected from different places in Serbia. Voucher specimens (BUNS) were used for the identification of species (Numbers: 2-0683, 2-0691, 2-0684, 2-0697 and 2-0690, respectively) [10]. The collected plant material was dried naturally, in small bouquets hanging upside down in shade at ambient temperature. When constant weight was obtained (after two weeks), only dry inflorescence was placed in a soft paper bag until further analysis.

Before the extraction, the raw material was grounded using a crusher (IKA, Multidrive B). The mean particle diameter (2 mm) of the ground plant material was determined by sieve analysis (*drogae minutim concisae*) [11].

## 2.2. Extract Preparation

### 2.2.1. Infusion

For infusion preparation, 2 g of ground plant material of each plant was added to 250 mL boiling distilled water and left to stand covered at room temperature for 20 min.

### 2.2.2. Maceration

Maceration was performed with 2 g ground plant material from each plant and 250 mL distilled water for 24 h at room temperature with occasional stirring.

### 2.2.3. Ultrasound-Assisted Extraction (UAE)

The UAE was performed with an ultrasonic bath with a volume of 3 L, frequency of 45 kHz and input power of ultrasound 410 W, using distilled water. Ground plant material weighting 2 g was extracted over 60 min at 60 °C with 250 mL distilled water.

### 2.2.4. Subcritical Water Extraction (SWE)

SWE was performed in an in-house-built subcritical water extractor (capacity 1.7 L) (described in detail in [12] (Figure 1). Extraction was done at 130 °C for 30 min at a heating rate of 10 °C/min with distilled water maintaining a sample-to-solvent ratio of 1:20 with an agitation rate of 3 Hz. Pressurization was done with nitrogen through the built-in lid valve (3), and was maintained as constant (30 bars) during the entire extraction process. After extraction, the vessel was cooled in a flow-through water bath at 25 °C and depressurized by valve opening.

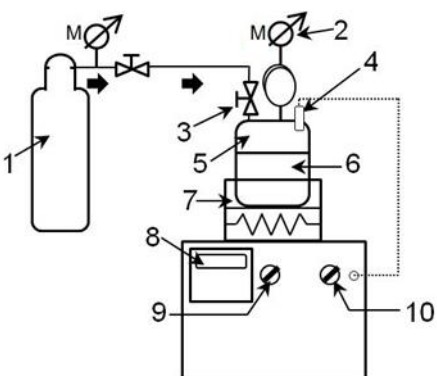

**Figure 1.** Schematic diagram of subcritical water extraction system: (1) nitrogen cylinder; (2) manometer; (3) input gas valve; (4) thermocouple for temperature measurement; (5) coverlid of extraction vessel; (6) extraction vessel; (7) vibrating platform; (8) digital temperature controller; (9) main switch; (10) switch for the vibrating platform.

## 2.3. Preliminary Chemical Characterization

All extracts were filtered (Syringe filter, 0.45 μm), evaporated to dryness by heating, and the extraction yield (dry extract (DE)) was calculated and expressed as a percentage of dry material. The obtained extracts were preliminarily chemically characterized in terms of total phenolic content (TPC) and total flavonoid content (TFC).

The TPC was determined by Folin–Ciocalteu method with some modifications [13]. In the presence of phenolic compounds, Folin–Ciocalteu reagent forms a blue-colored complex with a characteristic absorption maximum at 760 nm. Absorbance was measured using 1800 UV Vis spectrophotometer (Shimadzu, Kyoto, Japan). The total phenolic content was determined using gallic acid as a reference standard and the result was expressed as mg of gallic acid equivalents (GAE) per g of dry extract (mg GAE/g DE).

The TFC was determined spectrophotometrically based on complex formation of flavonoids with aluminum ions. The absorbance maximum was measured using 1800 UV Vis spectrophotometer at 460 nm [13]. The total flavonoid content was determined us-

ing quercetin as a reference standard and the result was expressed as mg of quercetin equivalents (QE) per g of dry extract (mg QE/g DE).

### 2.4. Statistical Analysis

Descriptive statistical analyses of the observed results were performed using one-way ANOVA (expressed as the mean $\pm$ standard deviation $-$SD), while the statistically significant differences were determined according to the post-hoc Tukey's honestly significant differences (HSD) test at $p \leq 0.05$ significance level. In addition, principal component analysis (PCA) enabled a more straightforward interpretation of the patterns of the investigated data by delivering details about specific variables that interact similarly and presenting them graphically. The data were explored using StatSoft Statistica 12 (StatSoft Inc., Tulsa, OK, USA).

## 3. Results

The results of the preliminary chemical characterization of the extracts obtained by conventional and non-conventional extraction methods are given in Table 1. As can be seen from this table, there is a statistically significant increase in DE content when applying SWE for all observed *Achillea* species. The highest value, 48.80 $\pm$ 1.76%, was observed for *A. asplenifolia*.

**Table 1.** Preliminary chemical characterization of *Achillea* sp. extracts obtained by conventional methods (infusion and maceration) and non-conventional methods (UAE and SWE).

| Sample | Species | Method | DE | TPC | TFC |
|---|---|---|---|---|---|
| 1 | *A. millefolium* | Infusion | 21.97 $\pm$ 0.46 [f,g] | 25.76 $\pm$ 3.92 [a] | 4.28 $\pm$ 0.21 [a] |
| 2 |  | Maceration | 12.38 $\pm$ 0.26 [a] | 30.01 $\pm$ 0.85 [a,b] | 5.44 $\pm$ 0.42 [a] |
| 3 |  | UAE | 19.71 $\pm$ 0.28 [d,e,f,g] | 25.71 $\pm$ 0.44 [a,b] | 5.18 $\pm$ 0.27 [a,b] |
| 4 |  | SWE | 41.50 $\pm$ 2.98 [i,j] | 93.63 $\pm$ 1.01 [a,b] | 6.24 $\pm$ 0.45 [a,b] |
| 5 | *A. clypeolata* | Infusion | 17.33 $\pm$ 1.28 [b,c,d,e] | 25.75 $\pm$ 7.76 [a,b] | 3.99 $\pm$ 0.04 [a,b] |
| 6 |  | Maceration | 13.11 $\pm$ 0.03 [a,b] | 29.05 $\pm$ 0.95 [a,b,c] | 7.54 $\pm$ 0.56 [a,b] |
| 7 |  | UAE | 17.48 $\pm$ 0.28 [b,c,d,e] | 34.70 $\pm$ 0.34 [a,b,c,d] | 9.76 $\pm$ 0.34 [b,c,d] |
| 8 |  | SWE | 45.30 $\pm$ 1.82 [j] | 77.61 $\pm$ 0.95 [b,c,d] | 4.78 $\pm$ 0.37 [c,d,e] |
| 9 | *A. asplenifolia* | Infusion | 22.24 $\pm$ 0.49 [f,g] | 32.26 $\pm$ 2.73 [b,c,d] | 4.64 $\pm$ 0.05 [d,e,f] |
| 10 |  | Maceration | 15.55 $\pm$ 0.40 [a,b,c,d] | 23.51 $\pm$ 0.77 [b,c, d] | 5.77 $\pm$ 0.49 [d,e,f,g] |
| 11 |  | UAE | 22.27 $\pm$ 0.32 [f,g] | 34.99 $\pm$ 0.48 [c,d] | 6.03 $\pm$ 0.13 [e,f,g] |
| 12 |  | SWE | 48.80 $\pm$ 1.76 [h,i] | 75.02 $\pm$ 0.18 [c,d] | 4.13 $\pm$ 0.33 [f,g] |
| 13 | *A. nobilis* subsp. *nelreichii* | Infusion | 23.63 $\pm$ 1.49 [g] | 35.93 $\pm$ 2.46 [c,d] | 6.79 $\pm$ 0.01 [g,h] |
| 14 |  | Maceration | 13.13 $\pm$ 0.43 [a,b] | 31.83 $\pm$ 0.29 [c,d] | 7.55 $\pm$ 0.29 [h,i] |
| 15 |  | UAE | 14.50 $\pm$ 0.19 [a,b,c] | 37.76 $\pm$ 0.42 [d] | 11.11 $\pm$ 0.22 [h,i] |
| 16 |  | SWE | 39.8 $\pm$ 4.7 [h,i] | 67.38 $\pm$ 0.37 [e] | 7.75 $\pm$ 0.29 [h,i] |
| 17 | *A. crithmifolia* | Infusion | 18.09 $\pm$ 0.69 [c,d,e,f] | 24.59 $\pm$ 6.94 [f] | 4.55 $\pm$ 0.09 [h,i] |
| 18 |  | Maceration | 14.32 $\pm$ 0.27 [a,b,c] | 31.77 $\pm$ 0.70 [g] | 6.53 $\pm$ 0.24 [i] |
| 19 |  | UAE | 20.16 $\pm$ 0.48 [e,f,g] | 34.59 $\pm$ 0.37 [h] | 7.55 $\pm$ 0.07 [j] |
| 20 |  | SWE | 35.70 $\pm$ 0.9 [h] | 90.12 $\pm$ 0.87 [h] | 7.38 $\pm$ 0.07 [k] |

TPC—Total Phenolic Content (in mg GAE/g DE), TFC—Total Flavonoid Content (in mg QE/g DE), DE—Dry Extract (in %). Different letters (a, b, c, d, e, f, g, h, i, j, k) printed in the same column show significantly different means of observed data ($p \leq 0.05$), according to post-hoc Tukey's HSD test.

For TPC, there is a statistically significant difference in content when applying different extraction methods for all observed *Achillea* species. The highest TPC values were observed after applying SWE, 93.63 $\pm$ 1.01 mg GAE/g DE for *A. millefolium*, and 90.12 $\pm$ 0.87 mg GAE/g DE for *A. crithmifolia*.

The results for TFC content in five different *Achillea* species using four different extraction methods revealed a statistically significant difference in values, with *A. nobilis*

subsp. *nelreichii* as the sample with the highest content of 11.11 ± 0.22 mg QE/g DE when using UAE as the extraction method.

According to Table 1, it could be concluded that the highest content of TPC and TFC were obtained by applying the non-conventional extraction methods UAE and SWE. On the other hand, when applying conventional extraction methods such as infusion and maceration, the highest values of TPC and TFC are noted in *A. nobilis* subsp. *nelreichii*.

Summarizing the results from Table 1, it could be concluded that SWE is the superior non-conventional extraction technique, and *A. nobilis* subsp. *nerleichii* presents as the most promising plant.

The PCA of the preliminary chemical characterization of *Achillea* sp. extracts obtained by conventional methods (infusion and maceration) and non-conventional methods (UAE and SWE) (Figure 2) showed that the first two principal components explained 97.15% of the total variance in the three parameters (DE, TPC and TFC). According to the results of the PCA, the content of DE (which contributed 50.67% of the total variance, based on correlations) showed a positive influence on PC1; also, the content of TPC (48.75%) exerted a positive effect on the PC1 coordinate. On the other hand, the content of TFC (93.96%) positively influenced the calculation of PC2 (Figure 2). Statistical analysis revealed evident differences between applied extraction methods, and the different cultivars tested. Higher concentrations of TPC and DE were obtained using SWE, while higher concentrations of TFC were noticed in samples extracted using UAE. The highest concentrations of TPC were noticed in *A. millefolium* and *A. crithmifolia* samples, while the highest concentrations of TFC were obtained in *Achillea nobilis* subsp. *nelreichii* and *A. clypeolata*. On the other hand, the highest concentrations of both TPC and TFC are detected in *Achillea nobilis* subsp. *nelreichii* as has already been noted in Table 1, and PCA confirmed these findings (Figure 2).

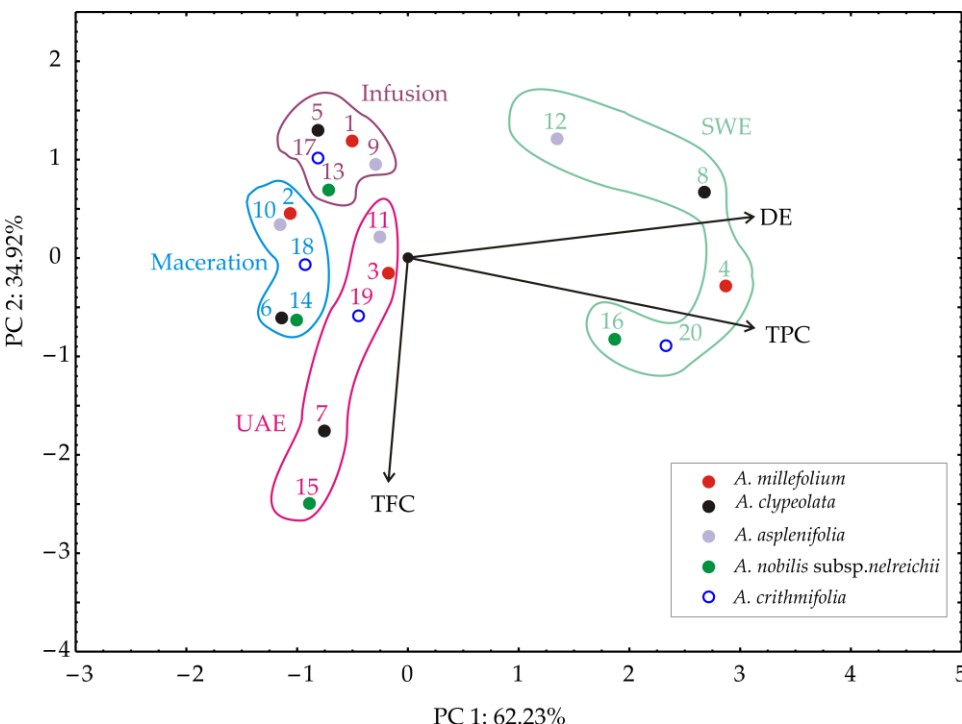

**Figure 2.** PCA biplot diagram depicting the relationships among *Achillea* samples regarding TPC, TFC and DE content.

## 4. Discussion

Phenolic compounds are plant secondary metabolites which the plant produces as a defense against stressful environmental conditions. These compounds are important as potential health-promoting constituents as part of functional diet. Also, they have huge

medicinal potential due to their antioxidant, anticancer, antibacterial, cardioprotective, anti-inflammation, immune system promoting and skin protective effects [14]. Although polyphenols exist in numerous plant materials, their quantity and type are dependent on different factors including cultivation practices, climatic conditions, morphology and soil composition, but also extraction technique and operational parameters [15,16].

Research has shown that *A. millefolium* is an important source of bioactive compounds, among them phenols and flavonoids, which makes it an essential plant in pharmacy and ethnobotany [17–19]. However, the number of studies addressing phenolic compound extraction from other *Achillea* species in the literature is scarce.

Investigation of the application of different solvents (natural deep eutectic solvents—NADES, 80% ethanol, 80% methanol and water) in the case of *A. millefolium* shows differences in TPC, antibacterial and antifungal properties [20]. Given that traditional medicine is mainly based on water extracts, this type of extract is more generally the focus of scientists. *A. millefolium* inflorescence infusion shows good antioxidant activity, confirmed antiphlogistic activity, as well as the potential to suppress lipopolysaccharide-induced inflammatory responses in murine macrophages [21].

Modern extraction techniques, especially in the case of value-added products, are very important from an environmental point of view. The valorization of *A. millefolium* by-products from the filter tea industry using SWE has shown that subcritical water can be successfully used for utilization of *A. millefolium* by-products for obtaining extracts rich in antioxidants [22]. Further, in the investigation of new substrates for kombucha beverages, TPC and TFC were quantified in infusion and SWE extracts of *A. millefolium* [16].

Considering the percentage of dry residue as a measure of extraction efficiency, for *Achillea* species, boiling water proved to be a better solvent than cold water. This observation is in line with the fact that heating increased solubility in water and also follows the recommendations of the European Medicines Agency (EMA) monograph for *A. millefolium*, which can be transferred to other similar *Achillea* species [23]. Furthermore, the content of phenolic compounds was approximately the same in the infusion and the macerate. However, the flavonoid content was slightly higher in the macerate, which may indicate potentially thermos-labile flavonoids in this species [24].

In respect to extraction yield and the content of total phenolic compounds, SWE showed itself to be a superior method in comparison to conventional techniques, but also in relation to ultrasonic extraction. The content of isolated flavonoids was relatively similar in all examined extracts. This may be caused by the limited water solvating capacity or by amount of herb material used for extraction. One study showed that the increase in sample-to-solvent ratio had negative influence on flavonoid content in SWE extracts [16].

Earlier literature data also indicated that SWE was a superior extraction approach in comparison to other techniques for phenol isolation [25], which was in accordance with the results obtained in this study. The most noticeable difference in the efficiency of isolation of phenolic compounds was seen for *A. millefolium*, where the total phenol content in the infusion and extract obtained by sonication was about 25 mg GAE/g DE, in the macerate about 30 mg GAE/g DE, while in the SWE extracts it was almost four times higher (93.63 mg GAE/g DE). Compared to other studies, the total phenol content in SWE extracts was higher than in organic extracts such as ethanol extract (71.33 mg GAE/g dw) or hexane/acetone extract (60.33 mg GAE/g dw). The TPC of water extract from same study was 13.96 mg GAE/g dw [26]. These results clearly indicate the superiority of SWE.

In addition to *A. millefolium* SWE extract, SWE extract from *A. crithmifolia* also had the highest total phenol content (90.12 mg GAE/g DE), while in other species this content was slightly lower (ranging from 67.38–77.61 mg GAE/g DE), but again almost twice as high as the content in other types of extracts.

Apart from this research, the results of numerous studies support SWE as one of the first choice modern methods. Study which compared ultrasound-assisted, solid–liquid and SWE of *Arctostaphylos uva-ursi* L. herbal dust indicated significantly better performance of the UAE and SWE for *A. uva-ursi* extracts in respect to higher extraction yield, TPCs and

TFCs [27]. The study which dealt with SWE of Indian ginseng noted that SWE showed considerably higher extraction yields for phenolics and flavonoids [28].

The main goal of this investigation was to determine which extraction method has the greatest potential for further in vitro and in vivo investigations. According to the preliminary results obtained in this study, SWE is superior in comparison to other extraction methods with water (infusion, maceration and UAE). This research indicates that SWE has great prospects for further examinations because of the significantly higher content of biologically active compounds (TPE and TFE), and could be suggested as the method of choice for extraction when dealing with *Achillea* species.

## 5. Conclusions

This research revealed that the highest TPC and TFC were delivered by applying non-conventional extraction methods such as ultrasound-assisted extraction and SWE. Higher concentrations of TPC were obtained using SWE, while higher concentrations of TFC were noticed in samples extracted using UAE. The highest concentrations of TPC were noticed in *A. millefolium* ($93.63 \pm 1.01$ mg QE/g DE) and *A. crithmifolia* ($90.12 \pm 0.87$ mg QE/g DE) samples, while the highest concentrations of TFC were obtained in *A. nobilis* subsp. *nelreichii* ($11.11 \pm 0.22$ mg QE/g DE) and *A. clypeolata* ($9.76 \pm 0.34$ mg QE/g DE). On the other hand, the highest concentrations in both total phenolic and TFC were detected in *A. nobilis* subsp. *Nelreichii,* highlighting it as a most promising plant for various applications.

**Author Contributions:** Conceptualization, N.G., J.Š.-G. and M.A.; methodology, J.Š.-G.; software, B.L.; validation, K.R. and T.B.-B.; formal analysis, T.B.-B.; investigation, K.R.; resources, B.Z.; data curation, B.L.; writing—original draft preparation, K.R.; writing—review and editing, J.Š.-G. and M.A.; visualization, B.L.; supervision, N.G.; project administration, M.A.; funding acquisition, M.A. All authors have read and agreed to the published version of the manuscript.

**Funding:** This research was funded by the Ministry of Education, Science and Technological Development of the Republic of Serbia, grant numbers 451-03-68/2022-14/200032, 451-03-68/2022-14/200134 and 451-03-68/2022-14/200114.

**Institutional Review Board Statement:** Not applicable.

**Informed Consent Statement:** Not applicable.

**Data Availability Statement:** Not applicable.

**Acknowledgments:** The authors gratefully acknowledge Nenad Stanojević, Adonis Ltd. Sokobanja, for their administrative and technical support, and Milica Rat, Curator of BUNS herbarium for help in determination of plant material.

**Conflicts of Interest:** The authors declare no conflict of interest.

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
