# Peer review of "Subcritical Water Extraction as an Effective Technique for the Isolation of Phenolic Compounds of Achillea Species"

_processes, doi:10.3390/pr11010086_

Round 1

Reviewer 1 Report

Abstract need to be revised. No clear details of the findings are mentioned in the abstract

How the plant materials are dried? No information provided.

Once UAE and SWE are abbreviated on their first mention no need to repeat the full forms every time

Results can be supported with any evidence as a supplementary material. Just providing a table and biplot diagram not enough to make the manuscript suitable to publish in a impact journals

Discussion can be modified. Instead of giving basic information, the obtained results and its related activity can be discussed better using literature

Conclusion can be revised further to highlight the major findings

Author Response

We would like to thank the Reviewers for taking the necessary time and effort to review the manuscript. We sincerely appreciate all your valuable comments and suggestions, which helped us improve the quality of the manuscript.

Author Response

We would like to thank the reviewers for their thoughtful comments and efforts towards improving our manuscript. 

Reviewer 3 Report

the manuscript is quite interesting for readers. however, several comments and suggestions are addressed to the author in order to improve the quality of the manuscript.

1. In line 20, please remove "the water"

2. in line 90, is the raw material grounded? how about the particle size of the material? is it standardize of all the raw materials?

3.  in my opinion, please add the schematic diagram of SWE

4.  is there any pressure on the SWE?

5 please add more discussion related the process of extraction

Author Response

Thank you for your work in revising our paper. 
